# Concentration Dependent Effect of Human Dermal Fibroblast Conditioned Medium (DFCM) from Three Various Origins on Keratinocytes Wound Healing

**DOI:** 10.3390/ijms21082929

**Published:** 2020-04-22

**Authors:** Manira Maarof, Shiplu Roy Chowdhury, Aminuddin Saim, Ruszymah Bt Hj Idrus, Yogeswaran Lokanathan

**Affiliations:** 1Tissue Engineering Centre, Faculty of Medicine, Universiti Kebangsaan Malaysia, Jalan Yaccob Latif, Cheras 56000, Kuala Lumpur, Malaysia; manira@ppukm.ukm.edu.my (M.M.); shiplu56@gmail.com (S.R.C.); ruszymah@ppukm.ukm.edu.my (R.B.H.I.); 2Ear Nose & Throat Consultant Clinic, Ampang Puteri Specialist Hospital, 1 Jalan Mamanda 9, Ampang 68000, Selangor, Malaysia; aminuddin_saim@yahoo.com; 3Department of Physiology, Faculty of Medicine, Universiti Kebangsaan Malaysia, Jalan Yaccob Latif, Cheras 56000, Kuala Lumpur, Malaysia

**Keywords:** keratinocytes, fibroblasts, dermal fibroblast conditioned medium, tissue engineering

## Abstract

Fibroblasts secrete many essential factors that can be collected from fibroblast culture medium, which is termed dermal fibroblast conditioned medium (DFCM). Fibroblasts isolated from human skin samples were cultured in vitro using the serum-free keratinocyte-specific medium (Epilife (KM1), or define keratinocytes serum-free medium, DKSFM (KM2) and serum-free fibroblast-specific medium (FM) to collect DFCM-KM1, DFCM-KM2, and DFCM-FM, respectively). We characterised and evaluated the effects of 100–1600 µg/mL DFCM on keratinocytes based on attachment, proliferation, migration and gene expression. Supplementation with 200–400 µg/mL keratinocyte-specific DFCM-KM1 and DFCM-KM2 enhanced the attachment, proliferation and migration of sub-confluent keratinocytes, whereas 200–1600 µg/mL DFCM-FM significantly increased the healing rate in the wound healing assay, and 400–800 µg/mL DFCM-FM was suitable to enhance keratinocyte attachment and proliferation. A real-time (RT^2^) profiler polymerase chain reaction (PCR) array showed that 42 genes in the DFCM groups had similar fold regulation compared to the control group and most of the genes were directly involved in wound healing. In conclusion, in vitro keratinocyte re-epithelialisation is supported by the fibroblast-secreted proteins in 200–400 µg/mL DFCM-KM1 and DFCM-KM2, and 400–800 µg/mL DFCM-FM, which could be useful for treating skin injuries.

## 1. Introduction

Epithelial cells such as keratinocytes play a role in re-epithelialisation via cell proliferation and migration in order to close wounds [1,2]. During healing, re-epithelialisation helps to prevent infection and excessive moisture loss [3] and to maintain body temperature [4]. In severe skin loss due to burns, chronic ulcers, pressure ulcers or other life-threatening conditions, split skin graft (SSG) is used as the gold standard of care. However, the SSG donor site can cause lesions equivalent to second- or even third-degree burns [5], and limited skin sources have led to the use of tissue-engineered skin substitutes.

Skin substitutes can be permanent or temporary, consisting of biomaterial matrix sheets containing autologous or allogenic cells [6]. Skin substitute implantation promotes wound healing via the secretion of mediators such as growth factors, chemokines, cytokines and extracellular matrix (ECM) [7,8]. However, with cellular-based skin substitutes, a longer time is required to obtain the desired number of cells, and this will therefore increase production costs. Besides, allogenic cells can induce an immune response that causes graft rejection [9].

The alternative approach to overcoming these shortcomings is a product with wound-healing mediators and extended shelf-life. Recently, the potential of conditioned medium (CM), i.e., the waste from cell culture medium containing secretory proteins, such as that from fibroblasts and mesenchymal stem cells, has been explored for wound healing. As CM is collected during cell culture, its bulk production and storage with or without delivery modality as an off-the-shelf product is feasible.

Dermal fibroblasts secrete growth factors and ECM (in soluble form) into the culture medium, which serve as natural, efficacious healing ingredients. Previously, we demonstrated that dermal fibroblast CM (DFCM) enhanced keratinocyte expansion in in vitro monolayer culture [10]. DFCM contains proteins that promote wound healing. The fibroblast culture medium is commonly supplemented with serum to support fibroblast growth. In contrast, keratinocytes are usually cultured in a serum-free medium containing growth supplements, as serum is known to induce the differentiation of keratinocytes, which stops its growth and proliferation [11,12]. Therefore, serum-free medium is preferred for the collection of the DFCM, which can be supplemented later into keratinocytes culture. In this study, we are using two serum- free keratinocytes specific medium from Gibco (Carlsbad, CA, USA), i.e., EpiLife™ Medium (referred to as KM1) and Defined Keratinocyte Serum-free Medium (DKSFM™) (referred to as KM2) that are widely used for culturing the keratinocytes with different growth supplement composition. The fibroblast-specific culture medium used in this study is F-12: Dulbecco’s Modified Eagle medium without serum; Sigma, St. Louis, MO, USA) (referred to as FM). This study aimed to evaluate the concentration-dependent effect of DFCM from different types of medium in promoting in vitro re-epithelialisation, based on keratinocyte attachment, proliferation, migration and gene regulation in wound healing.

## 2. Results

### 2.1. Effect of DFCM on Keratinocyte Attachment and Proliferation

The concentration-dependent effect of DFCM on keratinocyte attachment and proliferation was evaluated by supplementing 100–1600 µg/mL DFCM to keratinocyte-specific basal medium without growth supplement (KBM). KM1 (fresh serum-free keratinocyte-specific medium containing growth supplement) and KBM were used as the positive and negative controls, respectively. Keratinocytes cultured in KBM failed to attach onto the culture surface. However, the DFCM-supplemented KBM facilitated keratinocyte attachment with significantly higher efficiency than KBM only (Figure 1A,B). DFCM-KM1 and DFCM-KM2 supplementation resulted in similar keratinocyte attachment patterns, where cell attachment was highest with 100 µg/mL DFCM (DFCM-KM1, 2598.60 ± 384.44 cells/cm^2^; DFCM-KM2, 2604.83 ± 401.337 cells/cm^2^), and the values were close to that of the positive control (KM1). Higher DFCM-KM1 and DFCM-KM2 concentrations significantly decreased the keratinocyte attachment efficiency. In contrast, DFCM-FM (100 µg/mL) caused lower keratinocyte attachment (639.73 ± 466.4 cells/cm^2^) than DFCM-KM1 and DFCM-KM2. However, keratinocyte attachment was not significantly different, with higher DFCM-FM concentrations.

Figure 1C shows the concentration-dependent effect of DFCM on keratinocyte growth rate. The keratinocytes maintained their cobblestone or polygonal morphology in all DFCM and in the positive control even after three-day culture (Figure 1A). There was no growth when the keratinocytes were cultured in KBM. In contrast, the keratinocyte growth rate increased when DFCM concentrations increased, up until 400 μg/mL (DFCM-KM1 and DFCM-KM2) and 200 μg/mL (DFCM-FM); however, it decreased once the DFCM concentration exceeded the optimum concentration. The keratinocyte growth rate for all concentrations of DFCM-KM1 and DFCM-KM2 was comparable to that of the positive control, and was significantly higher at 400 μg/mL and 800 μg/mL DFCM-KM1 (400 μg/mL, 0.024 ± 0.002 per hour; 800 μg/mL, 0.022 ± 0.002 per hour). In comparison, supplementation with up to 200 μg/mL DFCM-FM led to a keratinocyte growth rate comparable to that of the positive control. However, the keratinocyte growth rate decreased sharply following supplementation with 800 µg/mL and 1600 µg/mL DFCM-FM, as compared to the positive control, i.e., DFCM-KM1 and DFCM-KM2. Immunocytochemical staining confirmed these results, where keratinocytes supplemented with 400 µg/mL DFCM-KM1 and 1600 µg/mL DFCM-KM2 had more proliferative cells, i.e., more Ki67 staining, compared to the control, while DFCM-FM supplementation resulted in fewer proliferative cells than the other groups (Figure 2A,B).

### 2.2. Effect of DFCM on Keratinocyte Migration

To evaluate the concentration-dependent effect of DFCM on cell migration, sub-confluent or confluent keratinocytes were supplemented with DFCM. The positive control was keratinocytes supplemented with complete medium, i.e., KM1; the negative control was KBM-supplemented keratinocytes. The DFCM-KM1–supplemented subconfluent keratinocytes showed comparable single cell migration rates to that of the control group (0.70 ± 0.04 μm/min); DFCM-KM2–supplemented cells had lower migration rates, whereas no concentration-dependent effect was observed for either DFCM-KM1 or DFCM-KM2 supplementation. In comparison, the keratinocyte migration rate decreased as DFCM-FM concentrations increased. At 100 μg/mL DFCM-FM, the keratinocyte migration rate was similar to that of the positive control KM1 (0.68 ± 0.05 μm/min), and decreased to 0.35 ± 0.02 μm/min at 1600 μg/mL DFCM-FM (Figure 3A,B). However, the in vitro wound healing rate in confluent keratinocytes increased with the DFCM-FM concentration up until 800 μg/mL DFCM-FM, and decreased slightly at 1600 μg/mL DFCM-FM. The wound healing rate following supplementation with 200–1600 μg/mL DFCM-FM was higher than that with DFCM-KM1, DFCM-KM2 and the control groups (Figure 4A,B). DFCM-KM1 and DFCM-KM2 also demonstrated concentration dependent effects, where the wound healing rate increased when concentrations increased up to 400 μg/mL, and decreased thereafter. At 200 and 400 μg/mL, the wound healing rate of DFCM-KM1 and DFCM-KM2 was similar to that of the control group, KM1.

Table 1 summarises the keratinocyte attachment, proliferation and migration results following DFCM-KM1, DFCM-KM2 and DFCM-FM supplementation. Based on the keratinocyte properties, 200–400 μg/mL DFCM-KM1 and DFCM-KM2 and 400–800 μg/mL DFCM-FM were effective for promoting in vitro re-epithelialisation.

### 2.3. Gene Expression Analysis

The role of the genes of interest in wound healing was analysed using gene expression profiling. Keratinocyte attachment, proliferation and migration with 200 μg/mL DFCM-KM1 and DFCM-KM2 and 400 μg/mL DFCM-FM were used in gene expression analysis. We analysed 84 genes, and cultured keratinocytes in all groups expressed 81 key genes central to the wound healing response; the three exceptions were interleukin-2 (*IL2*), cathepsin G (*CTSG*) and plasminogen (*PLG*). Forty-two genes had similar fold regulation in the DFCM groups than in the positive control (KM1); 23 genes had higher fold regulation in the DFCM groups than in the KM1 group; however, the difference was not significant. Table 2 shows the fold regulation of overexpressed genes in the DFCM groups, as compared to the KM1 group. Most of the genes expressed in the DFCM groups are involved directly in wound healing, ECM remodelling (*CTSV*, *PLAT*, *MMP9*, *MMP7*, *F13A1*, *SERPINE1*, *PLAU*, *TIMP1*), structural formation (*COL1A1*, *COL1A2*, *COL3A1*, *COL4A1*, *COL5A1*, *COL5A2*, *COL5A3*), growth factors (*FGF7*, *CSF3*, *TNF*, *CSF2*), inflammatory cytokines (*IL1B*, *CCL7*), cell adhesion (*ITGB5*) and signalling pathways (*WNT5A*). Sixteen genes were downregulated in the DFCM groups compared to the control group (Table 3), which may affect wound healing. For example, *COL14A1* (collagen type XIV alpha 1 chain) is involved in fibroproliferative scars, while *PTEN* is a tumour suppressor gene.

### 2.4. Pathway Interaction Analysis

All genes were examined using ingenuity pathway analysis (IPA) software to identify gene interaction networks. The comparison analysis for all DFCM generated 31 networks, with 11 overlapping networks. Table 4 lists the top networks for each group as generated by the IPA software with the highest scores, which showed network function such as cellular movement, growth and proliferation, cell to cell signalling and interaction. Figure 5, Figure 6 and Figure 7 show the single network for each DFCM, including the interaction of collagen (ECM), PDGF (platelet-derived growth factor) (growth factor) and integrin (cell adhesion molecules) that activates the ERK/MAPK (extracellular signal–regulated kinase/mitogen-activated protein kinase) pathway important for cellular signalling. The pathway interaction analysis also revealed the canonical pathways directly involved in wound healing, such as the integrin, EGF (epidermal growth factor), WNT/β-catenin and PI3K (phosphatidylinositol-4,5-bisphosphate 3-kinase catalytic subunit alpha) signalling pathways (Table 5).

## 3. Discussion

In recent years, research has focused on investigating proteins secreted by cells into culture medium, that are retrievable and useful for triggering the biological properties of wound healing [10,13]. Many studies, both in vitro and in preclinical models, have evaluated the potential of cell-secreted proteins in CM as substitutes for cellular-based therapies for wound healing [13,14,15]. However, little is known about the role of CM in re-epithelialisation. In native tissue, the re-epithelialisation mechanism depends on the role of fibroblasts and their interaction with keratinocytes and on the expression of secretory mediators such as growth factors, cytokines, chemokines, integrins, keratins and ECM [2,16,17]. The role of DFCM in keratinocyte proliferation and migration, which is required during wound healing re-epithelialisation, has been demonstrated [10]. Here, we demonstrate the concentration dependent effect of DFCM prepared using different fibroblast culture media on in vitro re-epithelialisation and gene regulation in wound healing.

There was a significant difference in the in vitro keratinocyte properties between the DFCM groups. The DFCM-KM1 and DFCM-KM2 groups had higher keratinocyte attachment and growth rates, because the medium used in their preparation was derived from KM1 and KM2, respectively, and include complete growth supplements (product data sheets of EpiLife and DKSFM (defined keratinocyte serum-free medium with supplement), respectively, Gibco). These may influence keratinocyte attachment and proliferation, as the growth factors aid cell attachment and proliferation (product data sheets of EpiLife and DKSFM, Gibco). In contrast, the medium used for DFCM-FM was purely basal medium without any growth supplement (product data sheet of F12:DMEM (Dulbecco’s modified Eagle’s medium), Sigma). Our previous study has revealed that supplementation of DFCM-FM to keratinocytes culture slightly decreases cell attachment and proliferation [10]. This might be due to the fact that FM medium itself is a specific medium to culture fibroblasts and it might not be an optimal culture medium to support keratinocyte growth. Besides, DFCM-FM contains a higher concentration of calcium as a potent inducer of differentiation, that slows the growth and proliferation of keratinocytes [12]. This calcium in DFCM-FM benefits cell differentiation and enhances cell migration in a cluster form.

Cell proliferation and migration are the key processes during wound healing, where the epithelium will cover the damaged collagen matrix or activate re-epithelialisation [18]. In the present study, between 100 and 1600 μg/mL DFCM supplementation was examined, to determine the effective dose for keratinocyte properties. The DFCM used to supplement into keratinocytes culture is the pooled DFCM from three biological samples of fibroblast. Keratinocytes were cultured with KBM only (0 μg/mL DFCM) as a negative control. Comparison with the negative control was performed to understand the role of DFCM in keratinocyte attachment, proliferation and migration. The results were also compared with the positive control KM1, i.e., keratinocyte growth medium (EpiLife containing growth supplements). Only a few cells survived and adhered on the culture, but failed to proliferate with KBM only. However, DFCM supplementation improved keratinocyte attachment (100 μg/mL DFCM-KM1 and DFCM-KM2; 400–800 μg/mL DFCM-FM) and keratinocyte growth rate (400–800 μg/mL DFCM-KM1 and DFCM-KM2; 100–200 μg/mL DFCM-FM) compared to the negative control, and the results were comparable with that of the positive control. The results correlate with the keratinocyte proliferation, where keratinocytes supplemented with 400 µg/mL DFCM-KM1 and 1600 µg/mL DFCM-KM2 had a higher percentage of proliferation marker (Ki67) compared to the positive control. Meanwhile, all DFCM-FM groups had fewer proliferative cells than the other groups. This was supported by our previous studies that also showed consistency with the results of this study, which showed that DFCM-FM improves cell migration but not cell attachment and proliferation [7,10,19]. 

In normal skin renewal, the epidermis can renew its multi-layered epithelium continuously, which involves new basal cell layer production and keratinocyte differentiation or terminal differentiation. While some basal cells divide actively, other cells begin to lose their nucleus and cytoplasmic organelles, which is associated with apoptosis. These cells will transform into the keratinised layer and then flake off from the skin surface [1]. However, we show that DFCM-supplemented keratinocytes maintain basal cell properties via the positive expression of cytokeratin 14. This indicates that the cells still actively proliferate and that DFCM does not inhibit cell growth.

However, higher DFCM concentrations do not improve in vitro keratinocyte properties. This is due to the presence of metabolic waste such as lactate, ammonia and amino acids, that have toxic effects that can retard cell growth [20,21]. Some studies have suggested that ammonia accumulation in culture medium may induce cytoplasm acidification, leading to apoptosis [22]. To remove the small components, we filtered the DFCM with a 3-kDa centrifugal filter, followed by dialysis with a 1-kDa cut-off membrane. Therefore, the suppression of keratinocyte proliferation may have been due to the effect of higher DFCM concentrations, which downregulated the expression of genes such as *FGF* (fibroblast growth factor), *EFG*, *PDGF* and *VEGF* (vascular endothelial growth factor), which are crucial in cell proliferation and migration during wound healing.

Although DFCM-KM1 was superior in terms of the resultant keratinocyte attachment, proliferation and migration, the 200–1600 μg/mL DFCM-FM groups showed better wound healing rates than the other groups. This is due to the paracrine stimulation of keratinocyte migration by fibroblast secretory proteins [7] and higher fibronectin concentration in the DFCM-FM, which enhances cell migration [21]. Fibronectin plays a major role in cell adhesion, growth, migration and differentiation, which is important in wound healing and embryonic development [16]. It is also important in regulating ECM such as collagen [23], fibrinogen [24] and laminin [21]. Fibronectin also regulates cell signaling, via interaction with other growth factors and proteins [25] for cell adhesion, growth and migration. Furthermore, DFCM-FM was prepared using a medium with a higher calcium concentration, which enhances cell migration [26].

Growth factors such as EGF, TGF-β (transforming growth factor beta), FGF, PDGF, IL-1, IL-6 and TNF-α (tumour necrosis factor alpha) accelerate re-epithelialisation [17,27], and some of them were present in the DFCM. Secretory factors such as EGF, TGF-α and PDGF, and the stimulation of signalling molecules such as tissue plasminogen activator, urokinase plasminogen activator and the matrix metalloproteinases (MMP-1, MMP-9, MMP-10) initiate cell migration [28,29]. Furthermore, cell adhesion molecules such as the β-integrins are important for stimulating cell migration and proliferation during re-epithelialisation [18,30].

An evaluation of gene regulation showed that gene upregulation was directly involved in wound healing in the DFCM groups, as compared to the control. The expression of functional genes for ECM remodelling, structural formation, growth factors, inflammatory cytokines, cell adhesion and the WNT signalling pathway is important during wound healing [31]. The remodelling enzymes, i.e., MMPs, involved in ECM degradation (i.e., involving collagen, casein, and laminin), typically have low activity, but it increases during tissue repair or the remodelling of diseased or inflamed tissues [32,33]. MMP activation increases collagen concentrations and fibril network formation, producing the more stable granulation tissue [31]. The results show the expression of MMP and growth factors such as FGF and TNF in the DFCM, which exert angiogenic activity, a key phase in wound healing [31,33]. In addition, the fold regulation of WNT signalling was also higher than that of the control. WNT is important for controlling cell proliferation, differentiation, migration and apoptosis, by activating the intracellular signalling cascades [34]. The pathway interaction analysis also showed MAPK pathway activation, which is important for integrating signalling or information for cellular growth, proliferation, differentiation, migration and carcinogenesis [35]. Our findings indicate that DFCM influences cell physiology by accelerating keratinocyte attachment, proliferation and migration, and that it has potential for initiating in vitro wound healing.

## 4. Materials and Methods 

Ethics Approval and Consent to Participate: All procedures performed in studies involving human participants were in accordance with the ethical standards of the responsible committee on human experimentation (UKM Research Ethics Committee [UKMREC]; UKM approval code 1.5.3.5/244/02-01-02-SF0964, approval date 14 June 2013) and with the 1964 Helsinki Declaration and its later amendments or comparable ethical standards. Informed consent was obtained from all patients included in the study.

### 4.1. Cell Isolation and Culture

Redundant skin tissue samples from abdominoplasty were obtained from three consenting healthy patients (*n* = 3) and were processed as described elsewhere [36]. The samples were digested with 0.6% type I collagenase (Worthington, NJ, USA) for 4–6 h, followed by 8–10 min cell dissociation using 0.05% trypsin-EDTA (Gibco, Carlsbad, CA, USA). The cells were then re-suspended in co-culture medium (equivalent mixture of EpiLife (Gibco) and F12:DMEM (1:1; FD; Sigma, St. Louis, MO, USA) supplemented with 10% fetal bovine serum (FBS; Gibco)) and seeded in 6-well culture plates (Greiner Bio-One, Monroe, NC, USA) at 37 °C in 5% CO_2_. The medium was replaced every 2–3 days. The fibroblasts were removed when the cells were 70–80% confluent and were sub-cultured in a T75 flask (Nunc, Rochester, NY, USA) using FD+10% FBS until passage 3 (P3).

### 4.2. Preparation and Collection of DFCM

P3 fibroblasts were used to prepare the DFCM. The culture medium from 80%–100% confluent fibroblasts was removed, and the cells were washed twice with Dulbecco’s phosphate-buffered saline (DPBS, Sigma-Aldrich, St. Louis, MO, USA. Then, fresh serum-free keratinocyte-specific medium with growth supplement (EpiLife^TM^; Gibco) (referred to as KM1), or defined keratinocyte serum-free medium with supplement (DKSFM; Gibco, Carlsbad, CA, USA) (referred to as KM2) or fibroblast-specific culture medium (F-12: Dulbecco’s Modified Eagle medium without serum; Sigma, St. Louis, MO, USA) (referred to as FM) was added to the fibroblasts separately. The cells were then incubated at 37 °C in a 5% CO_2_ incubator for 72 h, and the waste medium was collected and designated DFCM-KM1, DFCM-KM2 and DFCM-FM, respectively. The DFCMs were filtered using a 3-kDa Amicon Ultra-15 centrifugal filter (Merck Millipore, Frankfurter, Darmstadt, Germany) to concentrate the proteins and dialysed using a 1-kDa cut-off PlusOne Mini Dialysis Kit (GE Healthcare, Amersham, Buckinghamshire, UK). Protein concentration was determined using a bicinchoninic acid assay (BCA assay) (Sigma, St. Louis, MO, USA), and absorbance at 562 nm was measured using a spectrophotometer (Bio-Tek, Winooski, VT, USA). The protein quantity was estimated by comparison of the readings with those of protein standards (Sigma, St. Louis, MO, USA). 

### 4.3. Keratinocyte Biological Properties

#### 4.3.1. Keratinocyte Attachment and Proliferation

P3 keratinocytes were used to evaluate the effect of DFCM on keratinocyte biological properties. Serum-free keratinocyte basal culture medium without supplement (KBM) was supplemented with 100–1600 μg/mL DFCM-KM1, DFCM-KM2 and DFCM-FM. Keratinocytes were seeded at 1 × 10^4^ cells per cm^2^ in 12-well plates (Greiner Bio-One) and incubated at 37 °C in a 5% CO_2_ incubator mounted on a Nikon A1R-A1 microscope (Minato, Tokyo, Japan). Time-lapse imaging was captured for 72 h at 20-min intervals, in five random positions from each well. Three technical replicates were performed for each biological replicate (*n* = 3). The cell concentrations for attachment 24 h after seeding were determined as follows:(1)$$\mathrm{Cell}\;\mathrm{concentration}\;\left(\mathrm{cells}/{\mathrm{cm}}^{2}\right)=\frac{\mathrm{Total}\;\mathrm{number}\;\mathrm{of}\;\mathrm{cells}\;\mathrm{in}\;\mathrm{the}\;\mathrm{image}}{\mathrm{Area}\;\mathrm{of}\;\mathrm{the}\;\mathrm{image}\;\left({\mathrm{cm}}^{2}\right)}$$

The keratinocyte growth rate was calculated by measuring the cell concentrations after 24 and 72 h as follows:(2)$$\mathrm{Growth}\;\mathrm{rate}\;\left(h^{-1}\right)=\frac{Ln(\mathrm{Cell}\;\mathrm{concentration}\;\mathrm{at}\;72\;h/\mathrm{Cell}\;\mathrm{concentration}\;\mathrm{at}\;24\;h)}{48\;h}$$

#### 4.3.2. Immunocytochemical Staining

Keratinocytes were fixed with 4% paraformaldehyde (Sigma-Aldrich, St. Louis, MO, USA), permeabilised with 0.1% Triton X-100 solution (Sigma-Aldrich) and blocked with 10% goat serum (Sigma-Aldrich) for 1 h at 37 °C. Next, the cells were incubated with mouse cytokeratin 14 monoclonal antibody (Abcam, Cambridge, UK) and rabbit anti-human Ki67 antibody (Abcam) overnight at 4 °C, followed by incubation with goat anti-mouse immunoglobulin G (IgG) Alexa Fluor 488 (green fluorescent dye; Invitrogen, Waltham, MA, USA) and goat anti-rabbit IgG Alexa Fluor 594 (red fluorescent dye; Invitrogen) for 2 h at 37 °C in the dark. The cells were counterstained with DAPI (4′,6-diamidino-2-phenylindole, Dako, Glostrup, Denmark) and observed using a Nikon A1R-A1 confocal microscope (Nikon, Tokyo, Japan).

#### 4.3.3. Single Cell Migration

The single cell migration rate was analysed using NIS-Elements AR 3.1 imaging software (Nikon). P3 keratinocytes seeded at 1 × 10^4^ cells per cm^2^ were cultured with or without DFCM. The migration rate was calculated by measuring the distance travelled by the cells for 20 min, using the equation below. Each cell migration rate was calculated by averaging three 20-min segments (1 h in total). Three technical replicates were performed for each biological replicate (*n* = 3).
(3)$$\mathrm{Migration}\;\mathrm{rate}\;\left(µm/\min\right)=\frac{\sqrt{{\left(X_{2}-X_{1}\right)}^{2}+{\left(Y_{2}-Y_{1}\right)}^{2}}}{t_{2}-t_{1}}$$
*X*_1_: initial *x* coordinate, *Y*_1_: initial *y* coordinate, *t*_1_: initial time; *X*_2_: final *x* coordinate, *Y*_2_: final *y* coordinate, *t*_2_: final time

#### 4.3.4. Scratch Wound Assays

Confluent keratinocyte monolayers in the middle of each well were scratched with a sterile pipette tip. The culture medium was removed, and the cells were rinsed with DPBS (Sigma-Aldrich) and cultured in KBM supplemented with 100–1600 μg/mL DFCM. All scratch assays were performed in three technical replicates for each of three biological samples (*n* = 3). Live imaging was performed using a Nikon A1R-A1 confocal microscope to capture images at 20-min intervals, to quantify the wound-healing rate as follows:(4)$$\begin{array}{cc} In\;vitro\;wound & headling\left(µm^{2}/h\right)\\ & =\frac{\mathrm{Initial}\;\mathrm{area}\;\mathrm{of}\;\mathrm{the}\;\mathrm{wound}\;\left(µm^{2}\right)-\mathrm{Final}\;\mathrm{area}\;\mathrm{of}\;\mathrm{the}\;\mathrm{wound}\;\left(µm^{2}\right)}{\mathrm{Observation}\;\mathrm{time}\;\left(h\right)} \end{array}$$

### 4.4. Gene Expression Analysis

The expression of 84 genes crucial to the wound healing response was identified using a Human Wound Healing RT^2^ Profiler polymerase chain reaction (PCR) Array (catalogue number PAHS-121Z, Qiagen, The Netherlands). The keratinocytes were seeded in 6-well plates in four groups, with three biological replicates, and incubated at 37 °C in 5% CO_2_ until confluent. The scratch assay was performed and the cells were cultured for 48 h in KBM (DFCM-supplemented) and in KM1 with growth supplement (positive control). The best DFCM concentration for supplementation was chosen based on the effectiveness of keratinocyte attachment, proliferation and migration. Then, the keratinocytes were trypsinised, and RNA and complementary DNA (cDNA) were synthesized using a RNeasy Mini Kit and RT^2^ First Strand Kit (Qiagen, Hilden, Germany). The cDNA (with 1.67 μg RNA per group) and RT^2^ SYBR Green qPCR (quantitative PCR) master mixes (Qiagen) were then aliquoted into a RT^2^ profiler PCR array plate containing specific primers for the 84 genes. The reaction was run on a Qiagen Rotor-Gene Q real-time cycler, and the results were analysed using RT² Profiler PCR Array data analysis software (Qiagen).

### 4.5. Pathway Interaction Analysis 

Ingenuity pathway analysis (IPA) software can be used to analyse a targeted gene set, based on numerous enrichment and interaction relationships. The fold regulation for all DFCM groups, with gene symbols as the identifier, was imported into the IPA system (Ingenuity Systems, http://www.ingenuity.com) to analyse the gene interactions, including the canonical pathways and molecular interaction networks. The DFCM were compared using the comparison analysis function, with the setting limited to humans, and the top networks and canonical pathways generated were identified.

### 4.6. Statistical Analysis

The quantitative results are reported as the mean ± standard error of the mean (SEM). The results were analysed with two-way analysis of variance (ANOVA) with post-hoc Tukey Test, and the difference between groups was significant if *p* < 0.05.

## 5. Conclusions

DFCM shows potential as a supplement for enhancing in vitro re-epithelialisation. Supplementation with 200–400 µg/mL DFCM-KM1 and DFCM-KM2 enhances attachment, proliferation and migration in subconfluent keratinocytes. In contrast, 200–1600 µg/mL DFCM-FM enhanced the keratinocyte wound healing rate, while 400–800 µg/mL DFCM-FM enhanced keratinocyte attachment and proliferation. Many genes modulate wound healing and are beneficial for therapeutic application in the future. Our findings suggest that an effective dose delivery of DFCM is useful for enhancing re-epithelialisation. However, further investigation is required to demonstrate its efficacy in enhancing re-epithelialization in skin regeneration in vivo and its potential as a novel and promising alternative treatment for skin injuries.

## 6. Patents

Malaysian Patent Application No. PI 2017703287. Dermal Fibroblast Conditioned Sera Based on Defined Keratinocyte-Specific Medium for Skin Regeneration.

Malaysian Patent Application No. PI 2017703742. Dermal Fibroblast Conditioned Sera Based on Fibroblast-Specific Medium for Skin Regeneration.

## Figures and Tables

**Figure 1 ijms-21-02929-f001:**
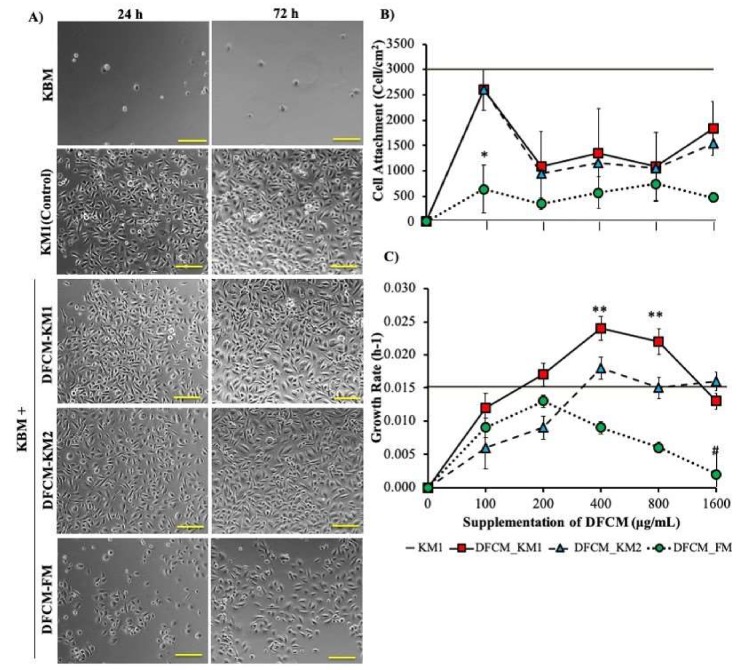
The effect of dermal fibroblast conditioned medium (DFCM) on keratinocyte attachment and proliferation. (**A**) Day 1 to day 3 morphology of keratinocytes supplemented with DFCM (100 μg/mL). The keratinocytes maintained their cobblestone or polygonal morphology after three-day culture. (**B**,**C**) The concentration-dependent effect of 100–1600 μg/mL DFCM on keratinocyte attachment (**B**) and growth rate (**C**). Positive control, KM1; negative control, KBM. * indicates significantly lower attachment with 100 μg/mL DFCM-FM as compared to DFCM-KM1 (*p* = 0.0009), DFCM-KM2 (*p* = 0.0009) and KM1 (*p* < 0.0001); ** represents a significantly higher growth rate, with 400 μg/mL and 800 μg/mL DFCM-KM1 supplementation as compared to 100 μg/mL and 1600 μg/mL DFCM-KM1, 100 μg/mL and 200 μg/mL DFCM-KM2, and 100 μg/mL and 400–1600 μg/mL DFCM-FM (*p* < 0.05); # represents a significantly lower growth rate than that for DFCM and KM1 (positive control) (*n* = 3). Scale bar = 100 μm.

**Figure 2 ijms-21-02929-f002:**
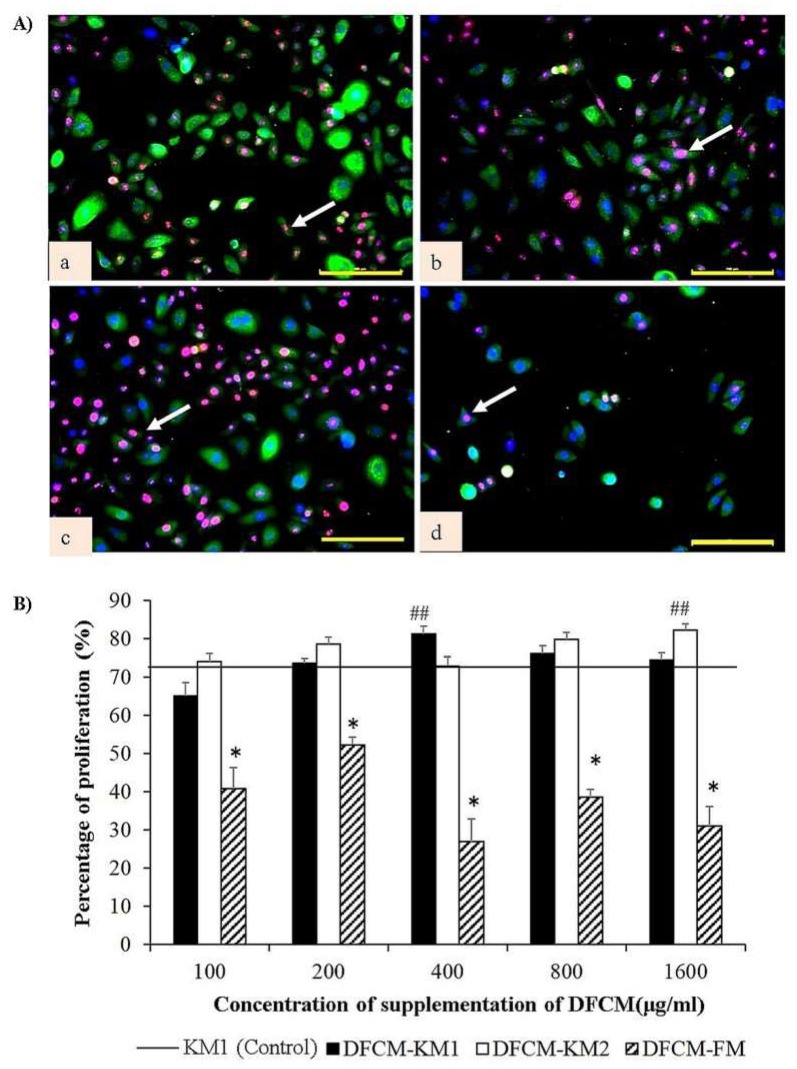
The effect of DFCM on keratinocyte proliferation. (**A**) Representative images of immunocytochemistry staining of keratinocytes supplemented with DFCM (100 μg/mL), with anti–cytokeratin 14 (green), anti-Ki67 (red) and nuclear staining (blue); (**a**) KM1 control, (**b**) KBM+DFCM-KM1, (**c**) KBM+DFCM-KM2, and (**d**) KBM+DFCM-FM. Arrow indicates positive expression of proliferative cell with anti-Ki67. Scale bar is 100 µm. (**B**) Quantitative evaluation (in percentage) of proliferative cells. Arrow shows representative cell with positive anti-Ki67 expression. ## represents significantly more proliferative cells in the DFCM group than in the control; * represents significantly fewer proliferative cells than in the other groups (*p* < 0.05) (*n* = 3). Scale bar = 100 μm.

**Figure 3 ijms-21-02929-f003:**
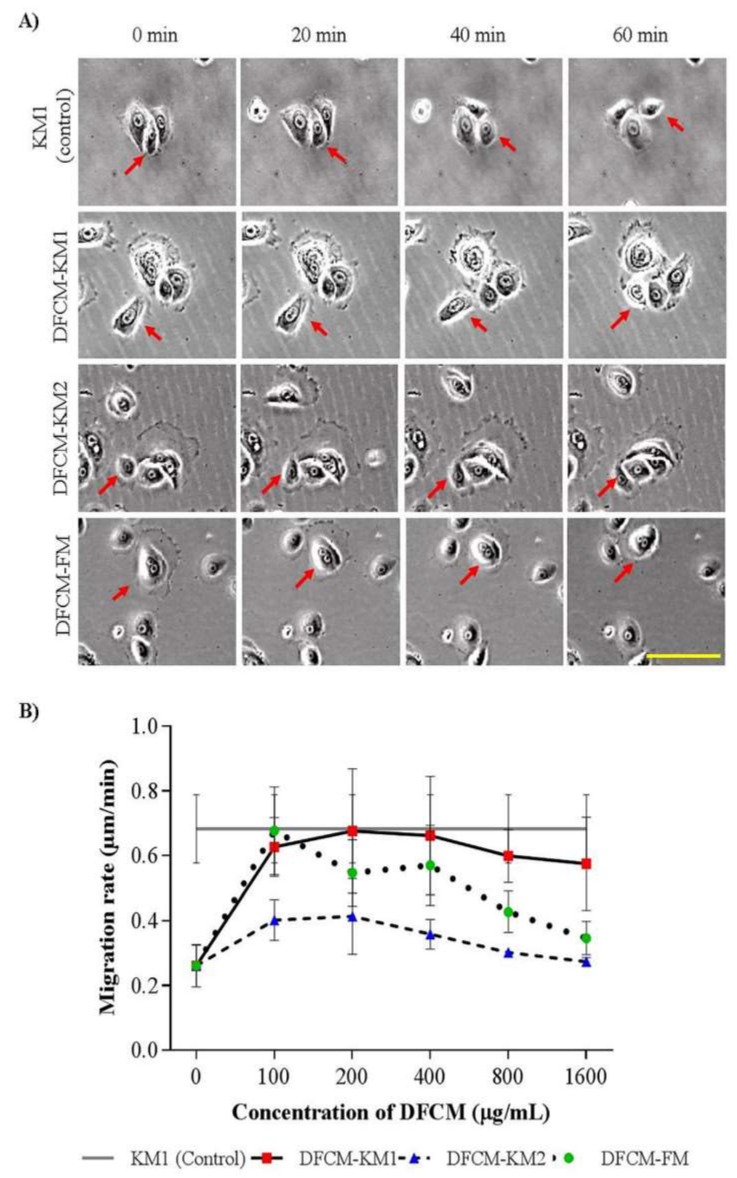
The effect of DFCM on keratinocyte migration at subconfluent state. (**A**) The images of keratinocyte migration supplemented with 100 µg/mL DFCM. Arrow indicates cell movement. Scale bar is 100 µm. (**B**) Keratinocyte migration rate. No concentration-dependent effect was observed in the DFCM-KM1 and DFCM-KM2 groups; DFCM-FM group showed a decreasing trend with increased DFCM-FM concentration. (*n* = 3). Scale bar = 100 μm.

**Figure 4 ijms-21-02929-f004:**
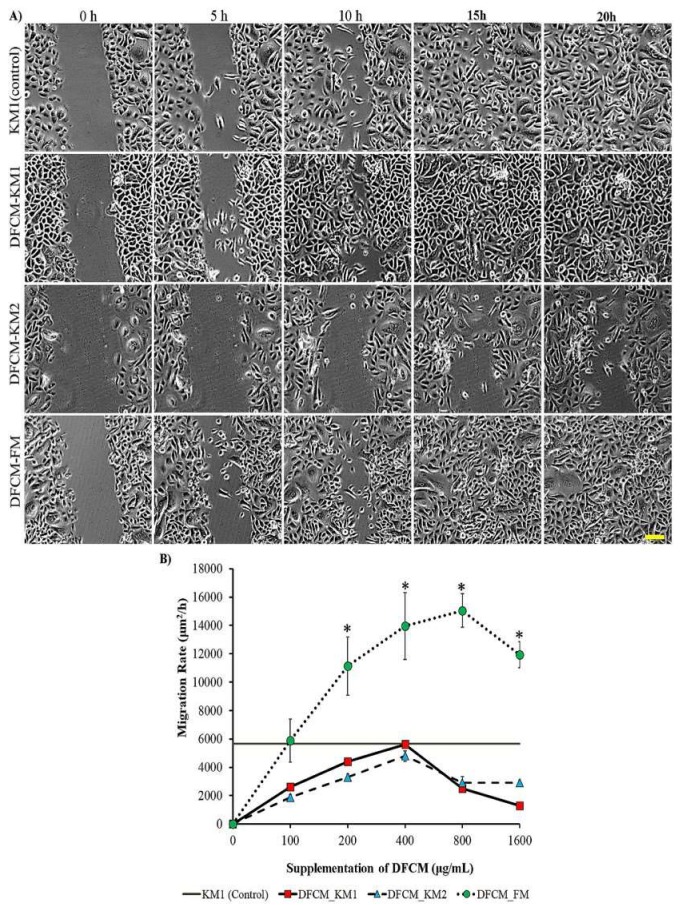
The effect of DFCM on keratinocyte wound healing. (**A**) The images of keratinocyte scratch assay supplemented with 400 µg/mL DFCM; scale bar is 100 µm. (**B**) Migration rate for wound healing assay. Here, * represents significantly higher migration rate in the DFCM-FM group than in the other groups (*p* < 0.05) (*n* = 3). Scale bar = 100 μm.

**Figure 5 ijms-21-02929-f005:**
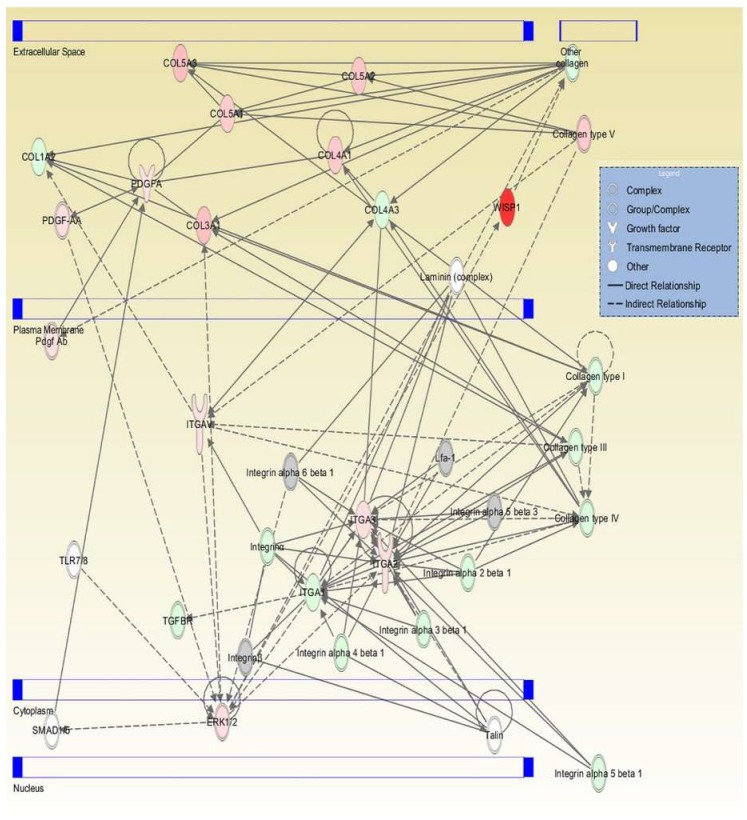
Graphic represents the interaction relationships for DFCM-KM. The mutual genes in the target network are displayed as nodes, and include collagen (ECM), PDGF (platelet-derived growth factor) (growth factor) and integrin (cell adhesion molecules), which activate the ERK/MAPK (extracellular signal–regulated kinase/mitogen-activated protein kinase) pathway. Red nodes indicate upregulated genes; green nodes indicate downregulated genes.

**Figure 6 ijms-21-02929-f006:**
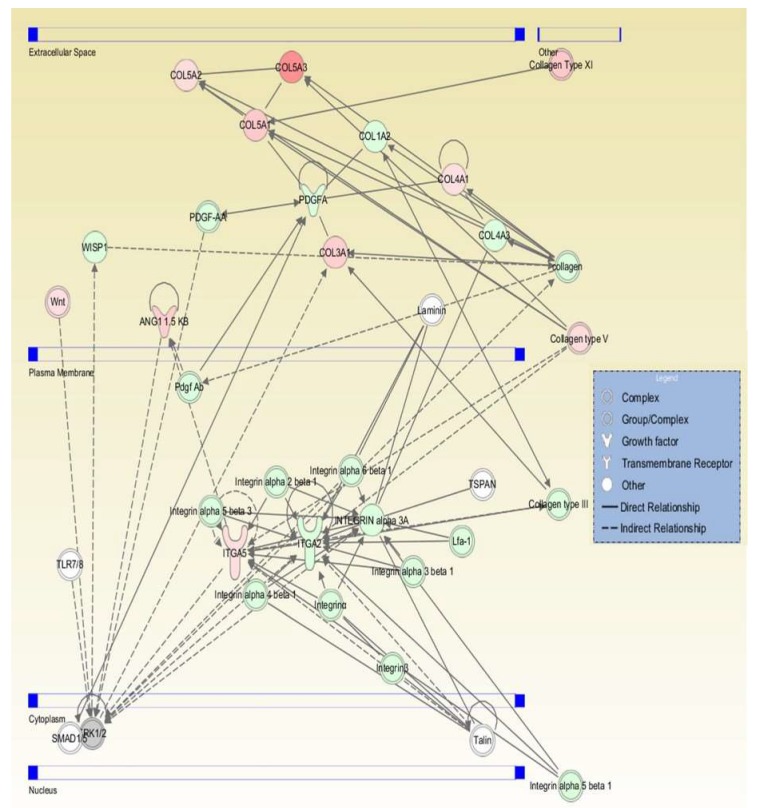
Graphic represents the interaction relationships for DFCM-KM2. The mutual genes in the target network are displayed as nodes, and include collagen (ECM), PDGF (growth factor) and integrin (cell adhesion molecules), which activate the ERK/MAPK pathway. Red nodes indicate upregulated genes; green nodes indicate downregulated genes.

**Figure 7 ijms-21-02929-f007:**
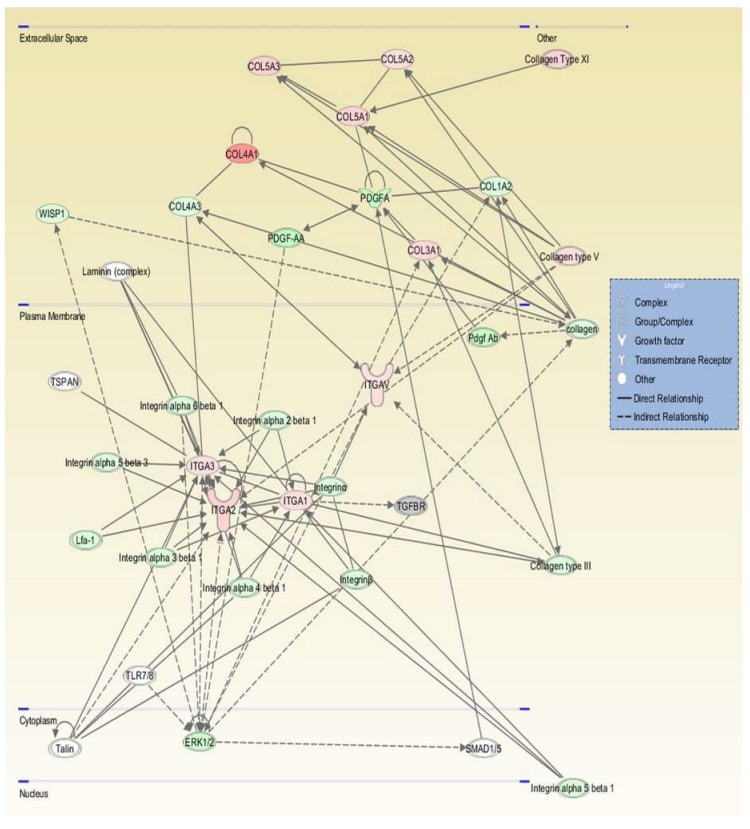
Graphic represents the interaction relationships for DFCM-FM. The mutual genes in the target network are displayed as nodes, and include collagen (ECM), PDGF (growth factor) and integrins (cell adhesion molecules), which activate the ERK/MAPK pathway. Red nodes indicate upregulated genes; green nodes indicate downregulated genes.

**Table 1 ijms-21-02929-t001:** Summary of concentration dependent effect of DFCM on keratinocytes properties.

Concentration (µg/mL)	DFCM-KM1	DFCM-KM2	DFCM-FM
**Attachment**			
100	+++	+++	+
200	+	+	+
400	++	++	+
800	+	+	+
1600	++	++	+
**Proliferation/Growth Rate**			
100	++	++	++
200	+++	++	+++
400	+++	+++	++
800	+++	+++	+
1600	++	+++	+
**Migration Rate**			
100	+	+	+++
200	++	++	++++
400	++	++	++++
800	+	+	++++
1600	+	+	++++

+, Significantly lower than control (<50%); ++, significantly lower than control (50–80%); +++, no significant difference compared to control; ++++, significantly higher than control. The highlighted grey areas show the effective concentration of DFCM.

**Table 2 ijms-21-02929-t002:** Up-regulated genes in DFCM groups as compared to KM1 group (control).

Functions	Gene	DFCM-KM1	DFCM-KM2	DFCM-FM
Extracellular Matrix (ECM) Structural Constituents	*COL1A2*	8.08 ± 4.84	6.35 ± 1.13	3.77 ± 8.17
	*COL5A2*	5.05 ± 3.01	3.10 ± 1.02	2.26 ± 4.82
	*COL3A1*	5.72 ± 4.16	4.37 ± 3.09	3.43 ± 8.00
	*COL1A1*	3.50 ± 12.86	3.83 ± 2.26	11.55 ± 4.82
	*COL5A1*	4.60 ± 9.44	4.85 ± 2.34	4.05± 7.67
	*COL5A3*	5.96 ± 13.83	10.09 ± 11.92	4.13 ± 3.76
	*COL4A1*	4.79 ± 4.57	2.84 ± 0.23	9.6 ± 15.30
Extracellular Matrix (ECM) Remodelling Enzymes	*CTSV*	6.33 ± 2.66	2.72 ± 0.94	12.32 ± 10.53
	*MMP9*	4.89 ± 3.05	2.59 ± 2.01	8.62 ± 3.64
	*F13A1*	2.33 ± 5.30	6.16 ± 2.56	7.31 ± 4.94
	*PLAT*	1.71 ± 3.20	2.57 ± 1.81	1.01 ± 6.04
	*MMP7*	9.76 ± 8.12	4.56 ± 2.81	1.67 ± 2.36
	*SERPINE1*	4.24 ± 1.70	1.86 ± 0.59	2.26 ± 6.34
	*PLAU*	4.24 ± 0.85	2.06 ± 0.99	4.49 ±2.66
	*TIMP1*	3.59 ± 1.85	1.84 ± 1.15	1.70 ± 0.98
Growth Factors	*FGF7*	3.19 ± 0.83	7.32 ± 8.93	5.15 ± 9.10
	*CSF3*	0.18 ± 2.25	4.10 ± 0.26	2.08 ± 4.71
	*TNF*	9.52 ± 17.77	7.36 ± 4.63	1.08 ± 1.52
	*CSF2*	7.37 ± 8.66	3.79 ± 1.10	12.03 ± 3.42
Inflammatory Cytokines & Chemokines	*IL1B*	1.71 ± 2.48	6.49 ± 5.86	5.38 ± 4.40
	*CCL7*	0.54 ± 0.76	7.32 ± 10.69	5.15 ± 3.42
Cell Adhesion Molecules	*ITGB5*	5.25 ± 4.94	4.35 ± 4.37	4.29 ± 4.45
WNT signalling	*WNT5A*	1.13 ± 3.89	2.47 ± 0.89	0.85 ± 6.65

**Table 3 ijms-21-02929-t003:** Of down-regulated genes in DFCM groups as compared to KM1group (control).

Functions	Gene	DFCM-KM1	DFCM-KM2	DFCM-FM
Extracellular Matrix (ECM) Structural Constituents	*COL14A1*	−11.95 ± 14.97	−9.09 ± 10.86	−1.13 ± 12.35
	*COL4A3*	−3.04 ± 1.29	−1.62 ± 1.07	−1.85 ± 2.78
Extracellular Matrix (ECM) Remodelling Enzymes	*FGA*	−22.87 ± 25.87	−15.99 ± 39.45	−17.87 ± 27.61
Growth Factors	*FGF2*	−104.87 ± 9.65	−2.58 ± 4.93	−2.24 ± 5.79
	*EGF* (epidermal growth factor)	−12.04 ± 3.00	−2.59 ± 1.85	−1.97 ± 0.31
	*IGF1*	−6.25 ± 15.95	−1.26 ± 1.97	−11.02 ± 17.88
	*ANGPT1*	−12.34 ± 11.97	4.37 ± 1.70	−2.80 ± 1.97
	*PDGFA*	1.29 ± 1.66	−1.88 ± 0.52	−7.63 ± 2.06
	*EGFR*	1.17 ± 1.92	−1.69 ± 0.92	−3.39 ± 1.84
	*VEGFA*	−1.50 ± 0.03	−2.01 ± 2.47	1.26 ± 6.35
Inflammatory Cytokines & Chemokines	*CXCL2*	−6.13 ± 10.33	1.37 ± 2.30	−6.30 ± 0.25
	*CCL2*	−29.91± 38.60	1.54 ± 0.55	8.10 ± 14.67
	*CXCL1*	−3.44 ± 7.91	1.69 ± 3.35	−1.63 ± 1.90
	*IL6*	−5.31 ± 21.87	1.72 ± 3.78	3.99 ± 2.96
Other Signal Transduction Genes	*PTGS2*	−2.43 ± 0.89	−7.32 ± 15.22	−1.17 ± 3.34
Cytoskeleton Regulators	*ACTC1*	−1.83 ± 7.91	−2.03 ± 1.34	5.21 ± 5.04
Kinase	*MAPK3*	1.98 ± 1.01	1.09 ± 2.50	−6.90 ± 12.18
	*PTEN*	1.19 ± 3.25	−1.43 ± 2.91	−3.98 ± 14.9
Cell Adhesion Molecules	*ITGA6*	1.85 ± 0.91	−1.62 ± 2.27	−2.03 ± 17.92

**Table 4 ijms-21-02929-t004:** Networks generated by pathway interaction analysis.

DFCM	ID	Associated Network Function	Score	Focus Molecules
DFCM-KM1	1	Connective tissue disorders, organismal injury and abnormalities	24	13
	2	Organismal functions, organismal injury and abnormalities, tissue morphology	18	10
	3	Cellular movement, cellular growth and proliferation	15	9
	4	Organismal injury and abnormalities, cellular movement, cell–cell signalling and interaction	13	8
	5	Cellular development, cellular growth and proliferation, haematological system development and function	9	6
DFCM-KM2	1	Connective tissue disorders, organismal injury and abnormalities	25	13
	2	Tissue morphology, haematological system development and function, tissue development	20	11
	3	Cellular development, cellular growth and proliferation, organ development	18	10
	4	Organismal injury and abnormalities, organismal functions, tissue morphology	16	9
	5	Cellular movement, haematological system development and function, immune cell trafficking	13	8
DFCM-FM	1	Connective tissue disorders, organismal injury and abnormalities	24	13
	2	Organismal injury and abnormalities, haematological system development and function	18	10
	3	Cell–cell signalling and interaction, cellular movement, haematological system development and function	11	7
	4	Organismal injury and abnormalities	11	7
	5	Cell–cell signalling and interaction, embryonic development, cellular development	11	7

**Table 5 ijms-21-02929-t005:** Pathway interaction analysis -generated canonical pathways involved in wound healing. The *p*-values represent the significance of the pathway.

Canonical Pathway	*p*-Value		
	DFCM-KM1	DFCM-KM2	DFCM-FM
Granulocyte Adhesion and Diapedesis	1.58 × 10^−19^	4.97 × 10^−25^	1.3 × 10^−24^
Agranulocyte Adhesion and Diapedesis	2.86 × 10^−19^	5.51 × 10^−23^	1.43 × 10^−22^
Integrin Signalling	6.22 × 10^−15^	1.05 × 10^−9^	2.13 × 10^−16^
EGF Signalling	4.29 × 10^−6^	1.6 × 10^−3^	1.82 × 10^−3^
TGF-β	2.2 × 10^−4^	3.26 × 10^−2^	7.44 × 10^−2^
WNT/β-catenin Signalling	2.14 × 10^−4^	1.53 × 10^−2^	2.16 × 10^−3^
PI3K/AKT Signalling	3.38 × 10^−10^	1.43 × 10^−7^	3.38 × 10^−10^

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
