# Peer review of "Concentration Dependent Effect of Human Dermal Fibroblast Conditioned Medium (DFCM) from Three Various Origins on Keratinocytes Wound Healing"

_ijms, 2020, doi:10.3390/ijms21082929_

Round 1

Reviewer 1 Report

The study is fairly interesting as it confirms previous results showing that fibroblasts contribute to epidermis repair. The idea here is to propose that the dermal fibroblasts conditioned medium could be used to promote epidermal repair.

The rational of 2 conditions using dermal fibroblasts conditioned medium made out of complete serum free keratinocytes medium is not clear and very confusing. The most relevant control here would be keratinocytes cultured in complete KM1, but this condition was not tested in the gene analysis. Or maybe the 2 conditions to compare would have been keratinocytes treated with complete KM1 and keratinocytes treated with the DFCM-KM1, in order that a clear comparison can be done.

Concentration determination brings an important bias because some of the used media already contain proteins from the beginning.

Aren't keratinocyte attachment and proliferation induced by DFCM-FM overestimated in Table 1 ?

Besides, testing another fibroblast strain would bring consistency to the data.

It is very confusing to the reader that both fresh serum-free keratinocyte-specific medium containing growth supplement and fibroblast-conditioned Epilife medium are named KM1.

Author Response

RESPONSE TO REVIEWER’S COMMENTS

Manuscript ID             : ijms-758406
Type of manuscript     : Article
Title                            : Concentration Dependent Effect of Human Dermal Fibroblast Conditioned Medium (DFCM) on Keratinocytes Wound Healing

Authors                      : Manira Maarof, Shiplu Roy Chowdhury, Aminuddin Saim, Ruszymah Bt Hj Idrus, Yogeswaran Lokanathan *

Dear Editor/ Reviewers,

Thank you very much for the valuable comments and suggestions. We have made changes according to the given recommendations. Following are the point-to-point responses to the comments. All the changes were highlighted in the text with track changes.

Reviewer Comments

Reviewer 1- IJMS

The study is fairly interesting as it confirms previous results showing that fibroblasts contribute to epidermis repair. The idea here is to propose that the dermal fibroblasts conditioned medium could be used to promote epidermal repair.

  1. The rational of 2 conditions using dermal fibroblasts conditioned medium made out of complete serum free keratinocytes medium is not clear and very confusing.

 Thank you for your comment. The paragraph has been added to the introduction section.

“The fibroblast culture medium is commonly supplemented with serum to support fibroblast growth. In contrast, keratinocytes are usually cultured in a serum-free medium containing growth supplements as serum is known to induce differentiation of keratinocytes, which stops its growth and proliferation (11,12). Therefore, serum-free medium is preferred for the collection of the dermal fibroblast conditioned medium (DFCM), which later can be supplemented into keratinocytes culture. In this study, we are using two serum- free keratinocytes specific medium from Gibco (USA), i.e. EpiLife™ Medium (referred as KM1) and Defined Keratinocyte Serum-free Medium (DKSFM™) (referred as KM2) that are widely used for culturing the keratinocytes with different growth supplement composition. The fibroblast-specific culture medium used in this study is F-12: Dulbecco’s Modified Eagle medium without serum; Sigma, USA) (referred as FM).  This study aimed to evaluate the concentration-dependent effect of DFCM from different types of medium in promoting in vitro re-epithelialisation based on keratinocyte attachment, proliferation, migration and gene regulation in wound healing.” (Introduction section, page 2, line 60-72).

  1. The most relevant control here would be keratinocytes cultured in complete KM1, but this condition was not tested in the gene analysis. Or maybe the 2 conditions to compare would have been keratinocytes treated with complete KM1 and keratinocytes treated with the DFCM-KM1, in order that a clear comparison can be done.

The keratinocytes cultured in the complete KM1 medium was the control group in this study and it was also tested for gene analysis. However, the data for KM1 alone was not shown in the manuscript because the gene expression analysis for DFCM-KM1, DFCM-KM2 and DFCM-FM were presented as fold regulation that was compared to KM1 as the control. The DFCM results were already normalised using KM1 results and this has been mentioned in the results section2.3 Gene Expression Analysis (Page 10, line 193-197).

  1. Concentration determination brings an important bias because some of the used media already contain proteins from the beginning.

We agree with the reviewer’s comment. However, we assumed that the effect observed in the keratinocytes biological properties are due to the proteins secreted by the fibroblast in the DFCM based on the observation on the DFCM-FM supplemented keratinocytes. The supplementation of DFCM-FM, which is a basal medium that purely contains the secreted proteins, showed that it can induce cell migration, an important process during wound healing. On other hand, these secreted proteins of DFCM-FM negatively affected the keratinocytes attachment and proliferation compared to the positive control, KM1  

  1. Aren't keratinocyte attachment and proliferation induced by DFCM-FM overestimated in Table 1?. Besides, testing another fibroblast strain would bring consistency to the data.

Thank you for your comment. The summary of concentration dependent effect of DFCM-FM on the keratinocytes properties already revised accordingly in Table 1 page 9.  

Our previous study has revealed that supplementation of DFCM-FM to keratinocytes culture slightly decrease cell attachment and proliferation (10). This might be due to the fact that FM medium itself is a specific medium to culture fibroblasts and it might not be an optimal culture medium to support keratinocyte growth. Besides, DFCM-FM contains a higher concentration of calcium as a potent inducer of differentiation that slows the growth and proliferation of keratinocytes (12). This calcium in DFCM-FM benefits cell differentiation and enhance cell migration in a cluster form (Discussion section, page 19 line 353- 359).

 In this study, three technical replicates were performed for each biological replicate (n = 3). The DFCM used to supplement into keratinocytes culture is the pooled DFCM from three biological samples of fibroblast. Besides, our previous studies also showed consistency with the results of this study which showed that supplementation of DFCM-FM improves cell migration but not cell attachment and proliferation (7,10,20) (Discussion section, page 19, line 364-365 and page 20, line 377-379).

  1. It is very confusing to the reader that both fresh serum-free keratinocyte-specific medium containing growth supplement and fibroblast-conditioned Epilife medium are named KM1.

The clarification of the medium used already mention in the method section. The fresh serum-free keratinocyte-specific medium with growth supplement (EpiLifeTM; Gibco, USA) (referred as KM1), or defined keratinocyte serum-free medium with supplement (DKSFM; Gibco, USA) (referred as KM2) or fibroblast-specific culture medium (F-12: Dulbecco’s Modified Eagle medium without serum; Sigma, USA) (referred as FM) is the medium used to culture confluence fibroblasts for 72h for collection of dermal fibroblast conditioned medium named as DFCM-KM1, DFCM-KM2 and DFCM-FM, respectively. (Method section, 4.2 Preparation and collection of DFCM, page 21 line 452-457). 

Reviewer 2 Report

There have been previous data indicating the importance of fibroblasts and keratinocytes together in post-wound circumstances and there have been results indicating that there are factors that are critical to the integration of processes within a system that will support regrowth. However, to the best of my knowledge, this is the first indication of the specificity of support that fibroblasts can contribute to support keratinocyte proliferation in an invitro model of wound repair. The authors have the potential to provide a new approach to dealing specifically with in- situ wound repair.

good work. 

Author Response

Thank you very much for the valuable comments

Round 2

Reviewer 1 Report

The paper can be accepted in its present form.

Author Response

Thank you for your comment and suggestion.